# Elderly patients with cancer admitted to intensive care unit: A multicenter study in a middle-income country

Antonio Paulo Nassar Junior[1]*, Mariane da Silva Trevisani[1], Barbara Beltrame Bettim[1], Fernando Godinho Zampieri[2,3,4], José Albani Carvalho, Jr.[5], Amilton Silva, Jr.[6], Flávio Geraldo Rezende de Freitas[7], Jorge Eduardo da Silva Soares Pinto[8], Edson Romano[9], Silvia Regina Ramos[10], Guilherme Brenande Alves Faria[11], Ulysses V. Andrade e Silva[12], Robson Correa Santos[13], Edmundo de Oliveira Tommasi[14], Ana Paula Pierre de Moraes[15], Bruno Azevedo da Cruz[16], Fernando Augusto Bozza[17,18], Pedro Caruso[1,19], Jorge Ibrahin Figueira Salluh[17], Marcio Soares[17]

1 A.C. Camargo Cancer Center, São Paulo, Brazil, 2 ID'Or, Research and Education Institute, São Paulo, Brazil, 3 Research Institute, HCor—Hospital do Coração, São Paulo, Brazil, 4 Center for Epidemiological and Clinical Research, University of Odense, Odense, Denmark, 5 Hospital São Luiz—Rede D'Or, São Paulo, Brazil, 6 Hospital Alemão Oswaldo Cruz, São Paulo, Brazil, 7 Hospital Sepaco, São Paulo, Brazil, 8 Hospital Norte D'Or, Rio de Janeiro, Brazil, 9 HCor—Hospital do Coração, São Paulo, Brazil, 10 Hospital São Luiz—Unidade Assunção, São Bernardo do Campo, Brazil, 11 Hospital Oeste D'Or, Rio de Janeiro, Brazil, 12 Fundação Pio XII—Hospital de Câncer de Barretos, Barretos, Brazil, 13 Hospital Estadual Adão Pereira Nunes, Duque de Caxias, Brazil, 14 Hospital Israelita Albert Sabin, Rio de Janeiro, Brazil, 15 Hospital de Câncer do Maranhão Tarquínio Lopes Filho, São Luís, Brazil, 16 Instituto Nacional de Câncer—HC II, Rio de Janeiro, Brazil, 17 D'Or Institute for Research and Education, Rio de Janeiro, Brazil, 18 Instituto Nacional de Infectologia Evandro Chagas, Fundação Oswaldo Cruz, Rio De Janeiro, Brazil, 19 Discipline of Pulmonology, Heart Institute, University of São Paulo, São Paulo, Brazil

* paulo.nassar@accamargo.org.br

**Data Availability Statement:** All relevant data are within the paper and its Supporting Information files.

## Abstract

### Background

Very elderly critically ill patients (ie, those older than 75 or 80 years) are an increasing population in intensive care units. However, patients with cancer have encompassed only a minority in epidemiological studies of very old critically-ill patients. We aimed to describe clinical characteristics and identify factors associated with hospital mortality in a cohort of patients aged 80 or older with cancer admitted to intensive care units (ICUs).

### Methods

This was a retrospective cohort study in 94 ICUs in Brazil. We included patients aged 80 years or older with active cancer who had an unplanned admission. We performed a mixed effect logistic regression model to identify variables independently associated with hospital mortality.

### Results

Of 4604 included patients, 1807 (39.2%) died in hospital. Solid metastatic (OR = 2.46; CI 95%, 2.01–3.00), hematological cancer (OR = 2.32; CI 95%, 1.75–3.09), moderate/severe

**Funding:** This study was supported by the National Council for Scientific and Technological Development (CNPq), Carlos Chagas Filho Foundation for Research Support of the State of Rio de Janeiro (FAPERJ) and by departmental funds from the D'Or Institute for Research and Education.

**Competing interests:** Dr. Soares and Dr. Salluh are founders and equity shareholders of Epimed Solutions®, which commercializes the Epimed Monitor System®, a cloud-based software for ICU management and benchmarking. The other authors declare that they have no conflict of interest. Dr. Zampieri has received grant for an investigator-initiated clinical trial from Bactiguard®, Sweden, which is unrelated to the aspects of this work. These do not alter our adherence to Plos One policies on sharing data and materials.

performance status impairment (OR = 1.59; CI 95%, 1.33–1.90) and use of vasopressors (OR = 4.74; CI 95%, 3.88–5.79), mechanical ventilation (OR = 1.54; CI 95%, 1.25–1.89) and renal replacement (OR = 1.81; CI 95%, 1.29–2.55) therapy were independently associated with increased hospital mortality. Emergency surgical admissions were associated with lower mortality compared to medical admissions (OR = 0.71; CI 95%, 0.52–0.96).

## Conclusions

Hospital mortality rate in very elderly critically ill patients with cancer with unplanned ICU admissions are lower than expected a priori. Cancer characteristics, performance status impairment and acute organ dysfunctions are associated with increased mortality.

## Introduction

Very elderly critically ill patients (ie, those older than 75 or 80 years) are an increasing population in intensive care units (ICU) [1, 2]. However, there is much controversy on how these patients should be managed. Although selected patients seem to benefit from ICU admission, elderly patients tend to have more ICU rejections [3] and a policy of systematic ICU admission of these patients has no positive effect on both short and long-term outcomes [4].

Recent research has focused on the epidemiology of elderly patients to identify the clinical path they follow after the ICU admission. As expected, those studies have showed increased mortality rates when compared to younger patients. However, mortality rates in the range of 20–40% may be considered acceptable under such patients' conditions [5–9]. Nevertheless, the vast majority of studies in this population were performed in high-income countries [5–9], which raises concerns on the generalization of results to low-and, more particularly, middle-income countries where half of the world population live and demographic changes have been accelerated [10].

In parallel, a considerable improvement in outcomes of critically ill cancer patients was observed in recent years [11]. Despite these findings, cancer is still considered a condition predisposing denial of ICU admission in elderly critically ill patients [4]. However, as anticancer therapies become safer and more effective, a larger number of elderly individuals are being treated [12, 13]. Moreover, the increased indications of surgical and pharmacologic interventions have expanded the need for intensive care for monitoring after procedures or treatment-related complications. Although there is no evidence to manage elderly critically ill patients with cancer differently from patients without cancer, evidence supporting this premise is very scarce [14]. To bridge this gap, we performed a multicenter study in Brazil to describe clinical characteristics, outcomes and to identify factors associated with hospital mortality of a large cohort of patients aged 80 years or older with cancer admitted to ICUs.

## Methods

### Design and setting

This was a retrospective study on prospectively collected data from two databases. The first is from the Orchestra study [15], a multicenter study performed in 93 ICUs from 55 hospitals in several Brazilian states from January 2014 to December 2015, and the second, from the A.C. Camargo Cancer Center, a dedicated cancer center in São Paulo Brazil, with 50 ICU beds, from January 2011 to December 2017. A.C. Camargo Cancer Center Local Ethics Committees

(CAAE: 86761718.0.0000.5432) and the Brazilian National Ethics Committee (CAAE: 19687113.8.1001.5249) approved the study without the need for informed consent, since all data were fully anonymized before researches could access them.

## Patients

We included all patients aged 80 years or older with active cancer who were admitted to the participant ICUs during the study period. We excluded patients admitted after elective surgeries.

## Data collection

We retrieved patients' data from the Epimed Monitor System (Epimed Solutions, Rio de Janeiro, Brazil) [16] as in other analysis of the Orchestra study and from the local database from A.C. Camargo Cancer Center. Trained healthcare workers inserted all clinical data in both databases. All data were deidentified. We collected data on patients' sex and age, type of ICU admission (medical, elective or urgent surgical), cancer type (hematological, solid locoregional or metastatic), performance status before hospital admission (evaluated by Eastern Cooperative Oncology Group [ECOG] categorized as absent/minor impairment, ie, ECOG 0 or 1, or moderate/severe impairment, categorized as ECOG 2 to 4) [17], a modified Charlson Comorbidity Index (CCI), which did not take into account points related to cancer status, the Simplified Acute Physiology Score (SAPS 3), use of organ support during ICU stay (vasopressors, invasive mechanical ventilation and renal replacement therapy), ICU and hospital length of stay (LOS), ICU and hospital mortality.

The primary outcome was hospital mortality. There was no missing data on the outcome. Information on performance status was absent for 193 patients (4.2%). There was minimal (<1%) missing data on mechanical ventilation, vasopressors and renal replacement use. There was no missing data on type of cancer, type of admission and CCI. We did not perform any imputation on these missing data and performed complete-case analysis.

## Statistical analysis

This study was mainly descriptive of the population of interest. We did not perform neither sample size nor power calculations, instead we present all available data from the included patients. All data are presented as frequencies (percentages) for categorical variables and as means (standard deviations) for continuous variables. We used chi-square test of independence for categorical variables and independent samples t-test test for continuous variables to compare two groups. Our variable of interest was hospital mortality.

We performed a mixed effect logistic regression model, with ICU as a random-effect, with predefined covariates (type of admission, type of cancer, performance status, modified CCI and use of mechanical ventilation, vasopressors and renal replacement therapy during ICU LOS) to evaluate its association with hospital mortality. We evaluated collinearity among the variables included in the models by calculating the Variation Inflation Factor (VIF). Arbitrarily, we considered VIF $\geq$ 2 as a diagnostic of multicollinearity. Odds ratio (OR) and 95% confidence intervals (CI 95%) were calculated for all these variables. In order to validate the prediction of the model, we randomly split the data into train (70%) and validation samples (30%). Calibration was evaluated by plotting the actual observed event frequency against the average predicted probability for each decile of a population, and qualitatively assessing the deviation from a diagonal line. Model discrimination power was assessed with area under the receiver operator curve (AUC). We used R version 3.5.1 for all analysis with the following packages lme4, dplyr and ggplot2.

## Results

There were 4604 eligible patients in the two databases during the study periods (Fig 1). Out of them, 856 (18.6%) patients were 90 years or older; 22 (0.5%) were 100 years or older. A total of 980 (21.3%) of patients died in ICU and 1807 (39.2%) died in hospital. Patients who did not survive to hospital stay had more metastatic and hematologic tumors, were more commonly admitted for medical reasons, had worse performance status and a higher burden of comorbidities. On the other hand, age was not associated with higher hospital mortality. Deceased patients used more invasive mechanical ventilation, vasopressor and renal replacement therapies during ICU LOS. Patients who deceased also have longer ICU and hospital length-of-stays (Table 1).

Most patients had solid tumors (n = 4186, 90.9%). Prostate, colorectal, breast and lung were the most common site of solid tumors. Lung cancer was more common among patients who deceased, while prostate cancer was more common among patients who survived to hospital

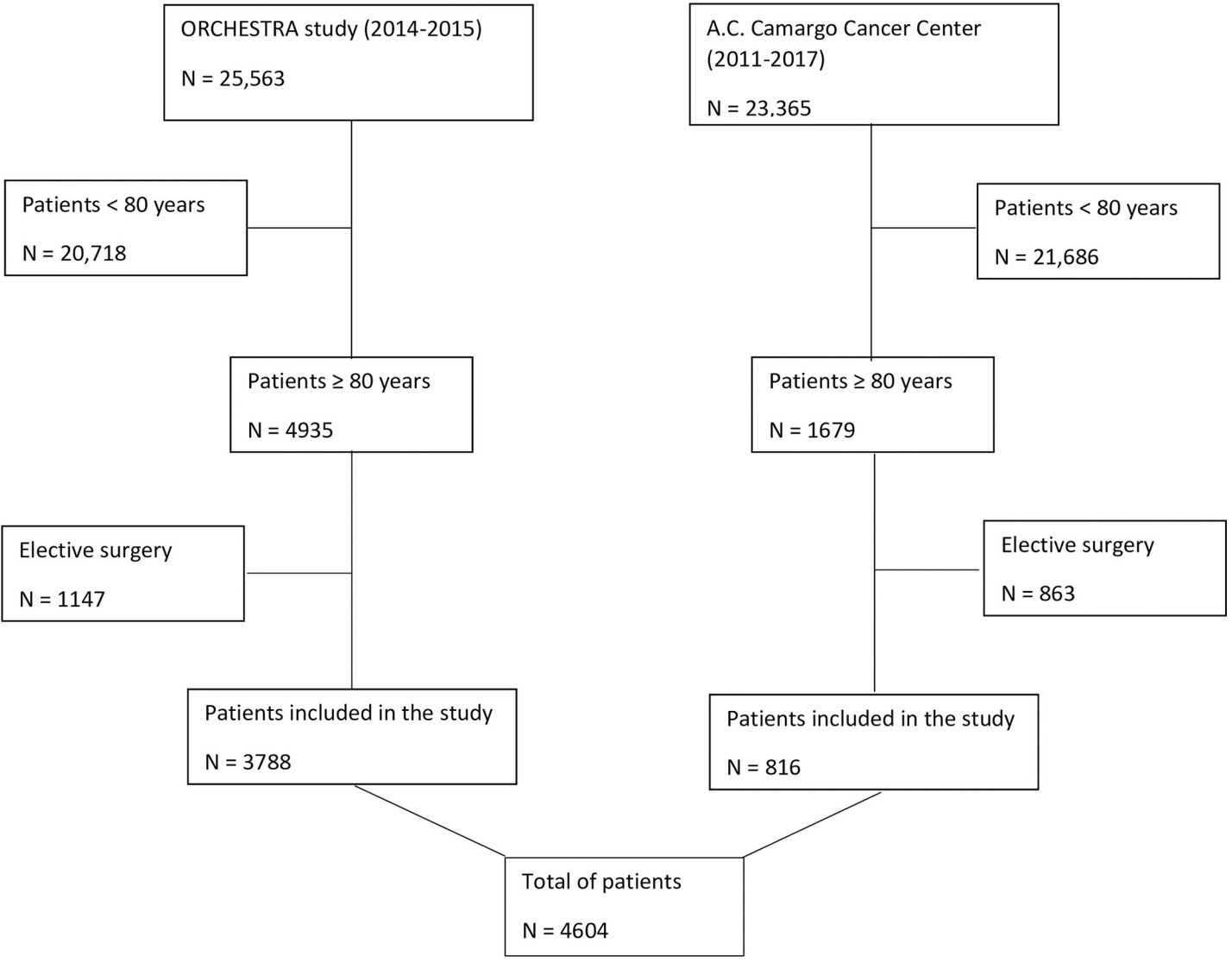

**Fig 1. Study flowchart.**

**Table 1. Patients' characteristics.**

| Variables | Alive (n = 2797) | Deceased (n = 1807) | p |
|---|---|---|---|
| Female sex, N (%) | 1349 (48.2) | 883 (48.9) | 0.67 |
| Age, years; mean (SD) | 85.6 (4.3) | 85.7 (4.3) | 0.29 |
| Type of cancer, N (%) | | | <0.01 |
| Solid, locoregional | 2027 (72.5) | 1001 (55.4) | |
| Solid, metastatic | 556 (19.9) | 602 (33.3) | |
| Hematological | 214 (7.7) | 204 (11.3) | |
| Type of admission, N (%) | | | <0.01 |
| Emergency surgery | 281 (10.0) | 134 (7.4) | |
| Medical | 2516 (90.0) | 1673 (92.6) | |
| Modified CCI, points; mean (SD) | 1.9 (1.1) | 2.0 (1.2) | 0.02 |
| Performance status, N(%)* | | | <0.01 |
| ECOG 0–1 | 1172 (41.9) | 597 (33.0) | |
| ECOG 2–4 | 1506 (53.8) | 1136 (62.9) | |
| Reason for admission, N (%) | | | <0.01 |
| Sepsis | 928 (33.2) | 711 (39.3) | |
| Cardiovascular | 421 (15.1) | 147 (8.1) | |
| Respiratory | 245 (8.8) | 267 (14.8) | |
| Neurological | 322 (11.5) | 167 (9.2) | |
| Renal/Metabolic | 141 (5.0) | 95 (5.3) | |
| Gastrointestinal | 246 (8.8) | 111 (6.1) | |
| SAPS 3, points; mean (SD) | 60.0 (11.8) | 70.0 (16.2) | <0.01 |
| ICU Complications, N (%) | | | |
| Vasopressor | 351 (12.5) | 695 (38.5) | <0.01 |
| Mechanical ventilation | 261 (9.3) | 793 (43.9) | <0.01 |
| Renal replacement therapy | 110 (3.9) | 207 (11.5) | <0.01 |
| ICU LOS, days; mean (SD) | 4.6 (5.7) | 8.0 (11.3) | <0.01 |
| Hospital LOS, days; mean (SD) | 19.1 (28.2) | 26.4 (57.2) | <0.01 |

CCI: Charlson Comorbidity Index. ECOG: Eastern Cooperative Oncology Group. ICU: Intensive Care Unit. LOS: Length of Stay. SAPS: Simplified Acute Physiology Score. SD: Standard Deviation.

*Data on ECOG was absent for 193 (4.2%) patients.

discharge. Among patients with hematological tumors (n = 418, 9.1%), lymphomas, leukemias and multiple myeloma were the most common type of tumors and occurred at similar patterns among patients who did and did not survive to hospital discharge (Table 2).

There was no multicollinearity among the predefined variables to be included in the logistic regression model (Table 3).

Metastatic cancer and hematologic cancer were independently associated with hospital mortality in comparison with locoregional solid tumors. Emergency surgical admissions were associated with lower hospital mortality than medical admissions. Performance status impairment, need for mechanical ventilation, vasopressors and renal replacement therapy were all also associated with increased hospital mortality (Fig 2). The model had a good discrimination (S1 Fig) power and was well calibrated (S2 Fig).

## Discussion

Our study showed that main clinical short-term outcomes for elderly critically ill cancer patients requiring ICU admission are reasonable. The observed mortality rates were

**Table 2. Type of tumors.**

| Solid tumors, N (%) | Alive (n = 2583) | Deceased (n = 1603) |
|---|---|---|
| Prostate | 559 (21.6) | 266 (16.6) |
| Colorectal | 342 (13.2) | 224 (14.0) |
| Breast | 386 (14.9) | 184 (11.5) |
| Lung | 213 (8.2) | 186 (11.6) |
| Head and neck | 181 (7.0) | 97 (6.0) |
| Renal | 164 (6.3) | 99 (6.2) |
| Stomach | 87 (3.3) | 66 (4.1) |
| Pancreas | 87 (3.3) | 74 (4.6) |
| Central nervous system | 81 (3.1) | 53 (3.3) |
| Liver and biliary tree | 71 (2.7) | 70 (4.4) |
| Hematological tumors, N ( | Alive (n = 214) | Deceased (n = 204) |
| Lymphoma | 87 (40.6) | 87 (42.6) |
| Leukemia | 58 (27.1) | 53 (26.0) |
| Multiple myeloma | 63 (29.4) | 50 (24.5) |

comparable to those reported in the literature for younger patients with cancer [11, 18] or undifferentiated elderly critically ill patients [9, 19]. In the present study, type of cancer, type of ICU admission, performance status and acute organ dysfunctions were all associated with increased hospital mortality.

Patients with cancer have been only a minority in large epidemiological studies of very old critically ill patients [5, 8]. Our study suggests elderly critically ill patients with cancer had a similar hospital mortality to those without cancer reported in previous studies [5, 7]. Use of vasopressors and renal replacement therapy in the patients of our study were similar to those of the studies of undifferentiated elderly critically ill patients [5–7], reinforcing similar severity among them.

Among baseline characteristics, performance status and type of cancer were associated with increased mortality. Performance status has widely been known as a prognostic factor in critically ill patients [17] and also, specifically, in elderly [20] and patients with cancer admitted to ICU [21]. As expected, the proportion of patients with at least moderate impairment of performance status in our study (60%) was much higher than that reported in studies of general critically ill patients [17]. However, the rates of performance status impairment found in our study are even higher than that found in some cohorts of elderly critically ill patients [22], suggesting that cancer burden may impact negatively on performance status of very old patients.

**Table 3. Variation inflation index of the selected variables to be included in the logistic regression model.**

| Variable | VIF |
|---|---|
| Type of admission | 1.018 |
| Type of cancer | 1.096 |
| Performance status | 1.008 |
| Mechanical ventilation | 1.374 |
| Vasopressors | 1.387 |
| Renal replacement therapy | 1.097 |
| Modified CCI | 1.112 |

CCI: Charlson Comorbidity Index. VIF: Variation Inflation Index.

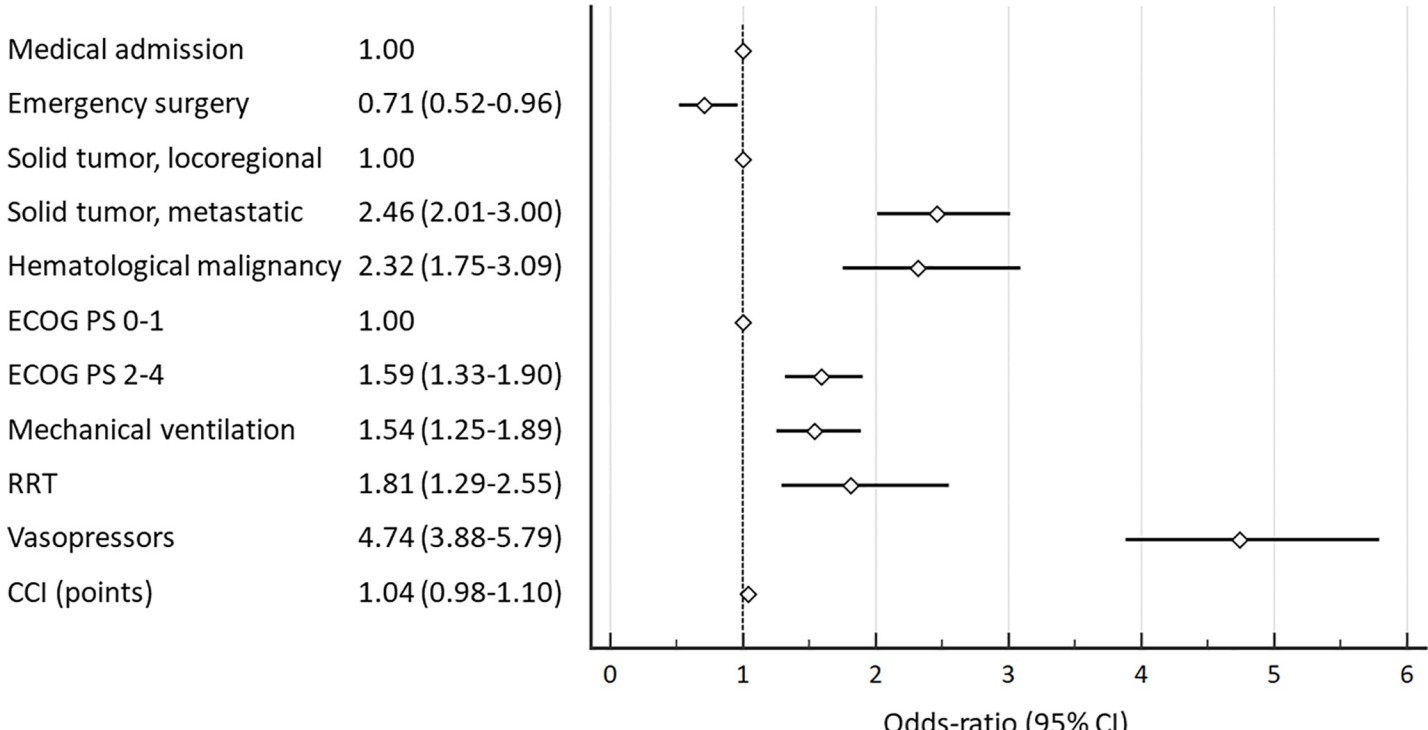

**Fig 2. Mixed effect logistic regression analysis for risk factors associated with hospital mortality.**

One specific finding of our study was that metastatic solid cancer and hematologic cancer were also associated with increased hospital mortality. Previous studies in critically ill patients with cancer have not consistently shown that type of cancer had such an impact [21]. It seems cancer status has a role only in mortality beyond 120 days [23, 24]. Our study suggests that, in a different manner from their younger counterparts, type of cancer has an impact on short-term outcomes of elderly critically ill patients. Therefore, performance status and type of cancer are two known characteristics at ICU admission which may be taken into account on decisions for care and on prognostic information of elderly critically ill patients with cancer.

Understanding the epidemiology and outcomes of elderly critically ill patients in middle-income countries is of paramount importance because these countries have been facing with a more accelerated demographic change than that faced by high-income countries in the end of 20th century [10]. Additionally, since rational resource utilization has been another challenge faced by middle-income countries, collaboration among intensivists, oncologists [25] and geriatricians [26] to the care of these patients will be fundamental.

Our study has some limitations. First, we could not assess frailty in our study patients. Frailty has been shown to be an important marker of mortality in critically ill, elderly and cancer patients [27]. Second, we also did not assess withholding or withdrawal of life-sustaining therapies. These decisions are obviously associated with increased mortality rates and may have had an impact on study patients' outcomes [28]. Third, we only have data on patients who were admitted to ICU. We cannot point out whether patients' which ICU admission was refused were more or less severely ill than those admitted neither they have or not worse outcomes. Finally, data on long-term survival and patient reported outcomes other than short-term mortality were unavailable.

Our study has also some strengths. It is a large multicenter cohort in middle-income country. Therefore, it adds information to those of high-income studies on elderly critically ill patients. Additionally, it sheds light on a population poorly described in the previous studies: elderly critically ill patients with cancer. Although these patients tend to have their ICU admission refused, it seems they also benefit from ICU admission in terms of short-term mortality. Whether this benefit is sustainable to long-term outcomes is current unknown.

In conclusion, elderly critically ill patients with cancer have comparable short-term outcomes after ICU admission to those of elderly patients without cancer reported in literature. Traditional markers of severity, such as medical admissions, use of life-sustaining therapies and performance status are also associated with increased mortality in elderly patients with cancer. In a different manner from their younger counterparts, metastatic solid cancer and hematologic cancer are associated with increased mortality in elderly critically ill patients and should be taken into consideration when planning care of these patients.

## Supporting information

**S1 Fig. Area under the receiver operator curve (AUC) for model calibration assessment.**
(DOCX)

**S2 Fig. Calibration curve plot.**
(DOCX)

**S1 Data.**
(XLSX)

**S1 File.**
(DOCX)

## Author Contributions

**Conceptualization:** Antonio Paulo Nassar Junior.

**Data curation:** Antonio Paulo Nassar Junior, Mariane da Silva Trevisani, Barbara Beltrame Bettim, Fernando Godinho Zampieri, José Albani Carvalho, Jr., Amilton Silva, Jr., Flávio Geraldo Rezende de Freitas, Jorge Eduardo da Silva Soares Pinto, Edson Romano, Silvia Regina Ramos, Guilherme Brenande Alves Faria, Ulysses V. Andrade e Silva, Robson Correa Santos, Edmundo de Oliveira Tommasi, Ana Paula Pierre de Moraes, Bruno Azevedo da Cruz, Pedro Caruso, Jorge Ibrahin Figueira Salluh, Marcio Soares.

**Formal analysis:** Antonio Paulo Nassar Junior, Barbara Beltrame Bettim.

**Funding acquisition:** Jorge Ibrahin Figueira Salluh, Marcio Soares.

**Investigation:** Antonio Paulo Nassar Junior, Mariane da Silva Trevisani, Pedro Caruso, Jorge Ibrahin Figueira Salluh, Marcio Soares.

**Methodology:** Antonio Paulo Nassar Junior, Barbara Beltrame Bettim, Fernando Godinho Zampieri, Marcio Soares.

**Project administration:** Jorge Ibrahin Figueira Salluh, Marcio Soares.

**Supervision:** Antonio Paulo Nassar Junior, Fernando Augusto Bozza, Jorge Ibrahin Figueira Salluh, Marcio Soares.

**Writing – original draft:** Antonio Paulo Nassar Junior, Amilton Silva, Jr., Jorge Eduardo da Silva Soares Pinto.

**Writing – review & editing:** Mariane da Silva Trevisani, Barbara Beltrame Bettim, Fernando
Godinho Zampieri, José Albani Carvalho, Jr., Flávio Geraldo Rezende de Freitas, Edson
Romano, Silvia Regina Ramos, Guilherme Brenande Alves Faria, Ulysses V. Andrade e
Silva, Robson Correa Santos, Edmundo de Oliveira Tommasi, Ana Paula Pierre de Moraes,
Bruno Azevedo da Cruz, Fernando Augusto Bozza, Pedro Caruso, Jorge Ibrahin Figueira
Salluh, Marcio Soares.

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
