## [Decision Letter · Decision Letter 0]

18 May 2020

PONE-D-20-05881

Very old patients with cancer admitted to intensive care unit: A multicenter study in a middle-income country

PLOS ONE

Dear Mr. Nassar Junior,

Thank you for submitting your manuscript to PLOS ONE. After careful consideration, we feel that it has merit but does not fully meet PLOS ONE’s publication criteria as it currently stands. Therefore, we invite you to submit a revised version of the manuscript that addresses the points raised during the review process.

We would appreciate receiving your revised manuscript by Jul 02 2020 11:59PM. To enhance the reproducibility of your results, we recommend that if applicable you deposit your laboratory protocols in protocols.io, where a protocol can be assigned its own identifier (DOI) such that it can be cited independently in the future. For instructions see: http://journals.plos.org/plosone/s/submission-guidelines#loc-laboratory-protocols

We look forward to receiving your revised manuscript.

Kind regards,

Valérie Pittet, PhD

Academic Editor

PLOS ONE

Journal Requirements:

'Local Ethics Committees and the Brazilian National Ethics Committee (CAAE:19687113.8.1001.5249 and CAAE: 86761718.0.0000.5432) approved the study without the need for informed consent.'

a.Please amend your current ethics statement to include the full name of the ethics committee/institutional review board(s) that approved your specific study.  

b.Once you have amended this/these statement(s) in the Methods section of the manuscript, please add the same text to the “Ethics Statement” field of the submission form (via “Edit Submission”).

3. In your ethics statement in the manuscript and in the online submission form, please provide additional information about the patient records used in your retrospective study. Specifically, please ensure that you have discussed whether all data were fully anonymized before you accessed them.

4. To comply with PLOS ONE submission guidelines, in your Methods section, please provide additional information regarding your statistical analyses. For more information on PLOS ONE's expectations for statistical reporting, please see https://journals.plos.org/plosone/s/submission-guidelines.#loc-statistical-reporting.

5. Please provide a sample size and power calculation in the Methods, or discuss the reasons for not performing one before study initiation.

6. For studies involving humans categorized by age authors should: 1) Explicitly describe their methods of categorizing human populations, 2) Define categories in as much detail as the study protocol allows, 3) Justify their choices of definitions and categories, 4) Explain whether (and if so, how) they controlled for confounding variables such as socioeconomic status, nutrition, environmental exposures, or similar factors in their analysis, and 5) Update outmoded terms and potentially stigmatizing labels to more current, acceptable terminology. Examples: “Very old” should be changed to "Elderly” etc.

7. Thank you for stating the following in the Competing Interests section:

'JIFS and MS are founders and proprietors of Epimed Solutions®. FGZ has received grant for an investigator-initiated clinical trial from Bactiguard®, Sweden, which is unrelated to the aspects of this work. The other authors report no conflicts of interest to declare.'

8. We note that you have stated that you will provide repository information for your data at acceptance. Should your manuscript be accepted for publication, we will hold it until you provide the relevant accession numbers or DOIs necessary to access your data. If you wish to make changes to your Data Availability statement, please describe these changes in your cover letter and we will update your Data Availability statement to reflect the information you provide.

Reviewers' comments:

Reviewer's Responses to Questions

**Comments to the Author**

1. Is the manuscript technically sound, and do the data support the conclusions?

Reviewer #1: No

Reviewer #2: Partly

Reviewer #3: Yes

2. Has the statistical analysis been performed appropriately and rigorously? 

Reviewer #1: No

Reviewer #2: Yes

Reviewer #3: Yes

3. Have the authors made all data underlying the findings in their manuscript fully available?

Reviewer #1: Yes

Reviewer #2: Yes

Reviewer #3: Yes

4. Is the manuscript presented in an intelligible fashion and written in standard English?

Reviewer #1: Yes

Reviewer #2: Yes

Reviewer #3: Yes

5. Review Comments to the Author

Reviewer #1: This is indeed an interesting question in the cancer setting. However, I have two major issues with the statistical analyses performed, and hence the discussion and conclusion.

1) If the outcome is mortality, why did you do logistic regression? Why did you not consider Cox regression and time to death?

2) The variables in the model may suffer from collinearity and hence it is recommended to create a directed acyclic graph to identify which variables should go into the model and how to estimate their independent effects on the outcome of interest.

Reviewer #2: This is an interesting piece of work with a few areas which could benefit from additional clarification;

How representative of the population as a whole is the population which is served by the two centres included in this study. Are the results scalable?

It is not entirely clear from the methods how confident the authors were with the completeness of variables such as cancer status which were used to identify the cohort.

How was the reason for admission to ICU defined?

There is no mention in the methods as to whether patients were on active treatment for their cancer at the time of admission to ICU, and if they were, what that was. This may be significantly impacting on the results of the study.

The message of the paper becomes a little confused in the discussion, the final paragraph of the paper is unclear. The focus appears to switch from being the impact of age on ICU mortality to being the impact of cancer on mortality in the very elderly. This could be made much stronger.

Reviewer #3: Very old patients with cancer admitted to intensive care unit: A multicentre study in a middle-income country.

The authors have performed a study aiming to describe clinical characteristics and identify factors associated with hospital mortality in a cohort of very old patients with cancer admitted to intensive care units.

The analyses are well performed and the only thing that makes me hesitate to recommend this paper for publication is the lack of information about the selection of patients to the study.

On page 3 line 78 the authors mention that cancer “is still considered a condition predisposing denial of ICU admission…”.

What kind of patients are in this study? They have cancer, but it’s hard to tell from the information given in the paper what cancer patients that are denied ICU admission. What are the characteristics for patients in the ICU without cancer?

Some of the cancers presented in Table2 (at least prostate cancer detected with a PSA-test) can be associated with a better health status than the background population https://zerocancer.org/learn/about-prostate-cancer/facts-statistics/.

If a screening program for breast cancer is present in Brazil, the same argument can be applied for breast cancer.

Why did the authors compare these patients to patients of similar age but without cancer?

6. PLOS authors have the option to publish the peer review history of their article (what does this mean?). If published, this will include your full peer review and any attached files.

Reviewer #1: No

Reviewer #2: No

Reviewer #3: No

---

## [Author Response · Author response to Decision Letter 0]

15 Jul 2020

Reviewer #1: This is indeed an interesting question in the cancer setting. However, I have two major issues with the statistical analyses performed, and hence the discussion and conclusion.

Dear reviewer,

Thank you for your time spent on peer-review.

1) If the outcome is mortality, why did you do logistic regression? Why did you not consider Cox regression and time to death?

We chose logistic regression because our aim was to evaluate factors associated with a specific event (i.e., hospital mortality). We did not intend to study a time to event outcome.

Additionally, a Cox regression model would not be appropriate for this study because some of the predefined variables (type of cancer, use of vasopressors and renal replacement therapy) included in the model did not respect the proportion hazards assumption according to the Schoenfeld residuals method, as can be seen in the figure in the attached file.

Therefore, in order to verify the association of the predefined covariates with the hospital mortality, we believe it would be more appropriate to perform a logistic regression model.

2) The variables in the model may suffer from collinearity and hence it is recommended to create a directed acyclic graph to identify which variables should go into the model and how to estimate their independent effects on the outcome of interest. 

We agree with the reviewer about collinearity. In the revised manuscript, We evaluated collinearity among the variables included in the models by calculating the Variation Inflation Factor (VIF). Arbitrarily, we considered, in a conservative approach, a VIF ≥ 2 as a diagnostic of multicollinearity. We added a sentence in Methods: “We evaluated collinearity among the variables included in the models by calculating the Variation Inflation Factor (VIF). Arbitrarily, we considered VIF ≥ 2 as a diagnostic of multicollinearity.” We also added the results of multicollinearity assessment in Results:

“There was no multicollinearity among the predefined variables to be included in the logistic regression model (Table 3).

Table 3. Variation inflation index of the selected variables to be included in the logistic regression model

Variable VIF

Type of admission 1.018

Type of cancer 1.096

Performance status 1.008

Mechanical ventilation 1.374

Vasopressors 1.387

Renal replacement therapy 1.097

Modified CCI 1.112

CCI: Charlson Comorbidity Index. VIF: Variation Inflation Index.”

Reviewer #2: This is an interesting piece of work with a few areas which could benefit from additional clarification;

Dear reviewer,

Thank you for your time spent on peer-review.

How representative of the population as a whole is the population which is served by the two centres included in this study. Are the results scalable?

Actually, this study was carried out with data from 56 centers. One of the databases is from a multicenter study with 55 hospitals (Orchestra study) and the other from a single large cancer center (A.C. Camargo Cancer Center). We believe the results are representative of Brazilian hospitals dedicated to care of patients with cancer.

It is not entirely clear from the methods how confident the authors were with the completeness of variables such as cancer status which were used to identify the cohort.

We agree with the reviewer that this information was not entirely clear. In the revised manuscript, we included the sentences: “There was no missing data on the outcome. Information on performance status was absent for 193 patients (4.2%). There was minimal (<1%) missing data on mechanical ventilation, vasopressors and renal replacement use. There was no missing data on type of cancer, type of admission and CCI. We did not perform any imputation on these missing data and performed complete-case analysis.”

How was the reason for admission to ICU defined?

In Orchestra database, reason for admission is extracted from a form fulfilled by a nurse or physician from the included ICU. In A.C. Camargo database, reason for admission is extracted from medical charts. The reason for admission is fulfilled by the intensivist in charge of the patient at the time of admission. In both databases, diagnoses are further categorized as sepsis, trauma or a disorder in an organ/system (neurologic, cardiovascular, respiratory, renal, gastrointestinal, surgical and others).

There is no mention in the methods as to whether patients were on active treatment for their cancer at the time of admission to ICU, and if they were, what that was. This may be significantly impacting on the results of the study.

It was an inclusion criteria for the study that patients were on active treatment for their cancer (In Methods, Patients, we highlighted this status in the revised manuscript: “We included all patients aged 80 years or older with active cancer who were admitted to the participant ICUs during the study period. We excluded patients admitted after elective surgeries.”). However, we don’t have, in neither database which were the treatment patients were on (for example, chemotherapy, target therapies, radiotherapy, etc).

The message of the paper becomes a little confused in the discussion, the final paragraph of the paper is unclear. The focus appears to switch from being the impact of age on ICU mortality to being the impact of cancer on mortality in the very elderly. This could be made much stronger.

We agree with the reviewer and apologize for not being clear. We believe we have two main messages. We have tried to emphasize both in the revised manuscript: 

1. Short-term mortality of elderly critically ill patients with cancer was not worse than that from elderly critically ill patients without cancer. Thus, in the second paragraph of discussion, we have written: “Our study suggests elderly critically ill patients with cancer had a similar hospital mortality to those without cancer reported in previous studies [5, 7]. Additionally, in conclusion, we have written “elderly critically ill patients with cancer have comparable short-term outcomes after ICU admission to those of elderly patients without cancer reported in literature.”

2. Type of cancer is associated with short-term mortality in elderly critically ill patients with cancer, what has not been consistently shown in previous studies in critically ill patients with cancer. In the fourth paragraph of discussion, we have written: “One specific finding of our study was that metastatic solid cancer and hematologic cancer were also associated with increased hospital mortality. Previous studies in critically ill patients with cancer have not consistently shown that type of cancer had such an impact[21]. (…) Our study suggests that, in a different manner from their younger counterparts, type of cancer has an impact on short-term outcomes of elderly critically ill patients.” Additionally, in conclusion, we have writen: “In a different manner from their younger counterparts, metastatic solid cancer and hematologic cancer are associated with increased mortality in elderly critically ill patients and should be taken into consideration when planning care of these patients.”

Reviewer #3: Very old patients with cancer admitted to intensive care unit: A multicentre study in a middle-income country.

The authors have performed a study aiming to describe clinical characteristics and identify factors associated with hospital mortality in a cohort of very old patients with cancer admitted to intensive care units.

Dear reviewer,

Thank you for your time spent on peer-review.

The analyses are well performed and the only thing that makes me hesitate to recommend this paper for publication is the lack of information about the selection of patients to the study.

On page 3 line 78 the authors mention that cancer “is still considered a condition predisposing denial of ICU admission…”.

What kind of patients are in this study? They have cancer, but it’s hard to tell from the information given in the paper what cancer patients that are denied ICU admission. What are the characteristics for patients in the ICU without cancer?

Thank you for the consideration. In the sentence mentioned previously, we had just tried to contextualize our study in the previous knowledge of critically ill patients. The study by Guidet et al (Guidet B, Leblanc G, Simon T, Woimant M, Quenot JP, Ganansia O, et al. Effect of Systematic Intensive Care Unit Triage on Long-term Mortality Among Critically Ill Elderly Patients in France: A Randomized Clinical Trial. Jama. 2017;318(15):1450-9.), which we cite after this sentence, aimed to evaluate whether a liberal ICU admission policy in elderly critically ill patients could reduce long-term mortality. However, elderly patients with cancer were excluded from this trial, possibly because it may be common sense that elderly patients with active cancer wouldn’t benefit from liberal ICU admission policies. Unfortunately, because of both databases are from patients ultimately admitted to ICU, we don’t have data on critically ill patients which ICU admission was denied or simply not considered.

Some of the cancers presented in Table2 (at least prostate cancer detected with a PSA-test) can be associated with a better health status than the background population https://zerocancer.org/learn/about-prostate-cancer/facts-statistics/.

If a screening program for breast cancer is present in Brazil, the same argument can be applied for breast cancer.

Why did the authors compare these patients to patients of similar age but without cancer?

We agree with the reviewer. In Brazil, we have a screening program for breast cancer too and maybe patients identified in screening programs may be associated with better health status. Anyway, our main aim was to describe a population of elderly critically ill patients with any type of cancer. Although patients with different types of cancer may have different long-term outcomes, we believe most healthcare personnel who take care of critically ill patients still see cancer as a unique variable. Additionally, unfortunately, we don’t have data on elderly patients without cancer from this study and can’t make such comparison.

---

## [Decision Letter · Decision Letter 1]

11 Aug 2020

Elderly patients with cancer admitted to intensive care unit: A multicenter study in a middle-income country

PONE-D-20-05881R1

Dear Dr. Nassar Junior,

We’re pleased to inform you that your manuscript has been judged scientifically suitable for publication and will be formally accepted for publication once it meets all outstanding technical requirements.

Kind regards,

Valérie Pittet, PhD

Academic Editor

PLOS ONE

Reviewers' comments:

Reviewer's Responses to Questions

**Comments to the Author**

1. If the authors have adequately addressed your comments raised in a previous round of review and you feel that this manuscript is now acceptable for publication, you may indicate that here to bypass the “Comments to the Author” section, enter your conflict of interest statement in the “Confidential to Editor” section, and submit your "Accept" recommendation.

Reviewer #2: All comments have been addressed

Reviewer #3: All comments have been addressed

2. Is the manuscript technically sound, and do the data support the conclusions?

Reviewer #2: Yes

Reviewer #3: Yes

3. Has the statistical analysis been performed appropriately and rigorously? 

Reviewer #2: Yes

Reviewer #3: Yes

4. Have the authors made all data underlying the findings in their manuscript fully available?

Reviewer #2: Yes

Reviewer #3: Yes

5. Is the manuscript presented in an intelligible fashion and written in standard English?

Reviewer #2: Yes

Reviewer #3: Yes

6. Review Comments to the Author

Reviewer #2: (No Response)

Reviewer #3: My concerns are satisfyingly addressed although it’s the problem of generalizability is still a concern. The potential reader of the paper has been presented with enough information to judge the generalizability. The paper is OK for publication.

7. PLOS authors have the option to publish the peer review history of their article (what does this mean?). If published, this will include your full peer review and any attached files.

Reviewer #2: No

Reviewer #3: No